# Developing a youth-friendly internet-enabled HIV risk calculator: A collaborative approach with young key populations, living in Soweto, South Africa

**Mamakiri Mulaudzi**[1,2,3]*, **Gugulethu Tshabalala**[1,2], **Stefanie Hornschuh**[2], **Kofi Ebenezer Okyere-dede**[4], **Minjue Wu**[5], **Oluwatobi Ifeloluwa Ariyo**[5], **Janan J. Dietrich**[1,2,6]

1 African Social Sciences Unit of Research and Evaluation (ASSURE), Wits Health Consortium, Faculty of Health Sciences, University of the Witwatersrand, Johannesburg, South Africa, 2 Perinatal HIV Research Unit (PHRU), School of Clinical Medicine, Faculty of Health Sciences, University of Witwatersrand, Johannesburg, South Africa, 3 School of Human and Community Development, Psychology, University of the Witwatersrand, Johannesburg, South Africa, 4 Dekode (Pty) Ltd, Software development company, Midrand, Johannesburg, South Africa, 5 Harvard Global Health Institute, Harvard University, Cambridge, Massachusetts, United States of America, 6 Health Systems Research Unit, South African Medical Research Council, Bellville, South Africa

* mamakiri.mulaudzi1@wits.ac.za, mamakirim@assureafrica.com

## 

**Data Availability Statement:** POPIA is now strictly enforced, and study participants did not agree to have their data made available publicly – even if

## Abstract

Although South Africa is the global epicenter of the HIV epidemic, the uptake of HIV testing and treatment among young people remains low. Concerns about confidentiality impede the utilization of HIV prevention services, which signals the need for discrete HIV prevention measures that leverage youth-friendly platforms. This paper describes the process of developing a youth-friendly internet-enabled HIV risk calculator in collaboration with young people, including young key populations aged between 18 and 24 years old. Using qualitative research, we conducted an exploratory study with 40 young people including young key population (lesbian, gay, bisexual, transgender (LGBT) individuals, men who have sex with men (MSM), and female sex workers). Eligible participants were young people aged between 18–24 years old and living in Soweto. Data was collected through two peer group discussions with young people aged 18–24 years, a once-off group discussion with the [Name of clinic removed for confidentiality] adolescent community advisory board members and once off face-to-face in-depth interviews with young key population groups: LGBT individuals, MSM, and female sex workers. LGBT individuals are identified as key populations because they face increased vulnerability to HIV/AIDS and other health risks due to societal stigma, discrimination, and obstacles in accessing healthcare and support services. The measures used to collect data included a socio-demographic questionnaire, a questionnaire on mobile phone usage, an HIV and STI risk assessment questionnaire, and a semi-structured interview guide. Framework analysis was used to analyse qualitative data through a qualitative data analysis software called NVivo. Descriptive statistics were summarized using SPSS for participant socio-demographics and mobile phone usage. Of the 40 enrolled participants, 58% were male, the median age was 20 (interquartile range 19–22.75), and 86% had access to the internet. Participants' recommendations were considered in

anonymised. Furthermore, due to the nature of the questions, which were sensitive in nature the full transcripts/qual data set cannot be shared, as it might expose participants inadvertently regardless of whether data is deidentified. Some participants shared personal stories, which were sensitive and were not included as part of the results of this manuscript. However, upon reasonable request parts of the data can be shared by request to hello@assureafrica.com / info@phru.co.za.

**Funding:** The work reported herein for MM was funded through the Female Academic Leaders Fellowship (FALF), the South African National Research Foundation (NRF), the Thuthuka PhD award, the National Institute of Humanities and Social Sciences, the South African Humanities Deans Association (SAHUDA), the Soweto Matlosana Centre for HIV/AIDS and TB (SoMCHAT)—through support from the South African Medical Research Council, and the Canada-Africa Prevention Trial Networks (CAPT Network). The data collection and analysis were supported by NRF, NIHSS, SoMCHAT, CAPTN. The publication of this research was supported by FALF. JJD was supported by the South African Medical Research Council through its Division of Research Capacity Development under the Early Investigators Programme from funding received from the South African National Treasury. The manuscript write up was support by South African Medical Research Council. SH was supported by the Consortium for Advanced Research Training in Africa (CARTA). CARTA is jointly led by the African Population and Health Research Center and the University of the Witwatersrand and funded by the Carnegie Corporation of New York (Grant No. G 19 57145), Sida (Grant No: 54100113), Uppsala Monitoring Center, Norwegian Agency for Development Cooperation (Norad), the Wellcome Trust [reference no. 107768/Z/15/Z], and the UK Foreign, Commonwealth & Development Office, with support from the Developing Excellence in Leadership, Training and Science in Africa (DELTAS Africa) programme. The manuscript write up was support by CARTA, Wellcome Trust, Sida and DELTAS Africa.

**Competing interests:** The authors have declared that no competing interests exist.

developing the HIV risk calculator. They indicated a preference for an easy-to-use, interactive, real-time assessment offering discrete and private means to self-assess HIV risk. In addition to providing feedback on the language and wording of the risk assessment tool, participants recommended creating a colorful, interactive and informational app. A collaborative and user-driven process is crucial for designing and developing HIV prevention tools for targeted groups. Participants emphasized that privacy, confidentiality, and ease of use contribute to the acceptability and willingness to use internet-enabled HIV prevention methods.

## Author summary

Despite medical advances in the treatment of HIV, the disease remains highly stigmatized, which is hampering the uptake of HIV prevention and treatment programs among South Africa's young people aged 15–24 years old. There are facilities available for HIV testing, treatment and management for young people in South Africa, however, barriers to privacy and confidentiality prevail. This study explored, through a collaborative process, how the development of a youth-friendly internet-enabled HIV risk calculator can help young people to overcome the hurdles they face in accessing HIV testing, prevention, and treatment, while affording them confidentiality and privacy, and providing valuable information about available services. While previous studies on mHealth for HIV prevention and risk assessment have focused primarily on MSM, this research addresses a gap by including young men and women of diverse sexual orientations and practices.

## Introduction

Over the years, there has been considerable progress made in the fight against the HIV epidemic [1]. However, the enduring disparities in infection rates serve as a clear sign that there is still work to be done and challenges to overcome. Although there have been innovations and treatment plans to curb the infection rates of HIV, South Africa is still the country with the highest infection rates globally, with approximately 7.6 million people living with HIV in 2022 [2]. Furthermore, in that same year, the HIV prevalence in South Africa accounted for approximately 19% of the global HIV infections [2]. Among individuals aged 15–24 years, an estimated 56,000 new HIV infections were reported in 2022 [2,3]. The HIV epidemic in South Africa is not homogeneous; rather, prevalence and incidence vary by age, race, gender, and socio-economic status, sexual orientation and are distinctly characterized at the local, district, and provincial levels [4,5]

Central to the HIV epidemic in South Africa are the key populations, including MSM, female sex workers, and LGBT individuals [6,7,8]. These marginalized groups continue to face a disproportionate burden of HIV transmission due to an intricate interplay of biological susceptibilities, social determinants, and structural barriers [9,10]. Their vulnerability to HIV infections often heightened by societal stigmatization, discrimination, and limited access to proper healthcare resources [9].

Studies collectively illuminate the intricate landscape of HIV risks across diverse populations in South Africa, highlighting the interplay of various contributing factors. Studies highlight a concern about increased HIV prevalence in populations such as MSM and female sex workers [6,8]. These findings show that the underlying reason behind these heightened HIV rates is transactional sex, along with instances of low and inconsistent condom usage and

difficulties in negotiating condom use [6,8]. This alarming blend of factors increases the susceptibility of these young key populations, thereby amplifying HIV transmission.

Mobile health (mHealth) has made significant strides in HIV prevention for young people across the world, including South Africa. mHealth involves using mobile phones and other wireless technologies to prevent, treat, manage and provide medical and public health care [11]. As smartphone ownership and use and internet accessibility have risen drastically among young people in South Africa [12], mHealth has enormous potential for HIV prevention. Internet-enabled applications (apps) are becoming increasingly viable platforms for HIV prevention for young South Africans [13]. Therefore, mobile phones and internet use for HIV risk assessment may become a highly acceptable strategy for this population group [14,15,16].

Involving young people in defining and designing interventions that work for them is crucial to providing age and culturally appropriate interventions [17]. Doing so encourages awareness among young people and promote access to these interventions [18].

Limited information is available on the use of mobile phones in HIV risk assessment and behavioral data collection among young people in South Africa [15]. Moreover, while many mobile apps have been developed for HIV prevention, there is limited evidence of the extent to which mHealth apps are being used and are acceptable by the target audience [19]. Despite the growing number of mHealth apps such as the WHO HTS Info, The Aspect™ HIVST, HIVS-MART; the usefulness of their content remains unknown [20,21,22,23]. A study by Raeesi et al. [24] aimed to validate some of these mHealth apps by rating the content of HIV-related mobile apps on the Google Play Store and Cafe Bazaar and determining the extent to which evidence-based medicine is incorporated into their content using a newly developed tool called the Evidence-Based Content Rating Tool of Mobile Health Applications (EBCRT-mHealth). According to the EBCRT-mHealth tool [24], one such app reviewed in the research is "WHO HTS Info" from the Google Play Store, which obtained a high rating of 5 (Excellent). The app was rated positively because it successfully delivered evidence-based information about HIV testing and services. Although the overall content quality of HIV-related apps was rated "poor" in 2018, it increased to "acceptable" in 2021 for Google Play Store apps. On the contrary, content quality in Cafe Bazaar apps declined in 2021, with content classified as "inappropriate" [24]. Developing an mHealth tool is a challenge but persuading the targeted group to use it presents an entirely new and considerably greater challenge [25,26].

Studies on mHealth for HIV prevention and HIV risk assessment have focused primarily on MSM, while overlooking other young population groups [27]. This research addresses a gap by including young men and women of diverse sexual orientations and practices. Furthermore, it describes the collaborative approach taken to develop a youth-friendly internet-enabled HIV risk calculator with young people aged 18–24 years living in Soweto, South Africa. The aim of this study is to describe the process taken to develop a youth-friendly internet-enabled HIV risk calculator for young people to assess their own risk for HIV acquisition and to identify their HIV risk factors.

## Methods

### Study population and setting

The study was conducted at the Perinatal HIV Research Unit (PHRU) based at Chris Hani Baragwanath Academic Hospital in Soweto, Johannesburg, South Africa. PHRU is affiliated with the University of the Witwatersrand and has been conducting research in Soweto for over 20 years. The unit has a walk-in HIV counseling and testing service (HTS) facility, accessible to anyone wanting to test for HIV. Participants lived in Soweto, a peri-urban area located in Gauteng, the country's most densely populated province, with approximately 11.3 million

people [28]. In 2017, HIV prevalence in the general population (15–49 years) in Gauteng was estimated at 17.6% [3]. A total of 46 young men and women participated in the study, including 23 in peer group discussions (PGDs), 18 in in-depth interviews (IDIs) and 5 Adolescent Community Advisory Board (ACAB) members in a group discussion. IDIs included young key populations: LGBT individuals, MSM, and female sex workers.

## Study design

To develop a youth-friendly internet-enabled HIV risk calculator collaboratively, we applied an exploratory qualitative approach, using peer group discussions, and in-depth interviews.

## Recruitment process

**Peer group discussion (PGD) participants.**   We performed active community recruitment in various locations in Soweto, such as community centers, shopping centers, and taxi ranks. Two trained fieldworkers approached young people, informed them about the study, and obtained contact details from those willing to participate to schedule them for a PGD. Of the 14 young women and 14 young men who were invited to the PGDs, 12 women and 11 men took part. To participate in the PGDs, they had to be aged 18–24 years, currently residing in Soweto, and able to read and write English.

**In-Depth interview (IDI) participants.**   We recruited IDI participants using purposive sampling and a chain referral strategy [29]—one of the most useful techniques to recruit marginalized individuals to access LGBT individuals, MSM and female sex workers [30]. Purposive sampling involved actively selecting participants who met specific inclusion criteria relevant to the study's objectives. We focused on individuals aged 18–24 years, identifying as LGBT, MSM, or female sex workers, currently residing in Soweto, being sexually active (i.e., having engaged in sexual intercourse in the past six months at the time of the study), and able to read and write English.

To complement the purposive sampling, we employed a chain referral sampling method. Chain referral sampling entailed asking each participant who completed an IDI to refer other young people who met the study's inclusion criteria. The initial contact with possible participants were recruited via active community recruitment where study fieldworkers went into the community to hand out pamphlets and where possible take contact details (cell phone number, age, gender and residence area) of individuals who showed interest to participate in the study. Following a completed IDI, participants were asked to refer other young people they knew who might meet the criteria for participation. Additionally, we used clinic-based recruitment through PHRU's HIV counselling and testing service facility and other ongoing projects to reach young key populations.

Young people who met the study criteria were approached for IDIs. Individuals interested in participating received an invitation flyer with the study's contact details. They could contact the study team either by visiting in person at the clinic, or via short messaging services (SMS), WhatsApp, or by sending a 'please call me' notification.

**Group discussion with Adolescent Community Advisory Board (ACAB) participants.**
Communication was sent to the Community Liaison Officers (CLOs) at the clinic to request participation from the ACAB in Soweto. At the time of the study, the ACAB comprised 14 members—five males and nine females—aged 16–24 years. Following approval from the CLOs, a fieldworker from our research team attended the ACAB meeting to provide information about the study and the purpose of involving the ACAB in the project. While eight ACAB members volunteered for the study, only five participated in the discussion.

## Data collection procedures

Fig 1 provides a representation of the data collection process, and a detailed description of each process is provided below.

**Peer group discussions.** Two trained female facilitators who were conversant in IsiZulu and Sesotho conducted two rounds of PGDs with young women and men. The first round of PGDs was conducted with a group of young women (n = 12) in the morning and a group of young men (n = 10) in the afternoon on the same day. All PGDs were conducted using participants' preferred languages, which were IsiZulu, Sesotho and English. Participants completed a paper-based self-administered, five-minute demographic questionnaire and a questionnaire on mobile phone usage followed by a 10-minute paper-based HIV and STI risk assessment questionnaire.

The participants were informed that the researchers were investigating ways to develop the questionnaire into a youth-friendly internet-enabled HIV risk calculator, which would be presented to them at their second PGD. Participants were asked to critically review the content, focusing on the type of questions presented, identifying any missing questions, discussing questions they found difficult to respond to, and providing suggestions on how to reformulate those questions.The second round of PGDs with the same group of young women and men was conducted about one week later to seek young people's opinions about how the HIV and STIs risk assessment questionnaire could be developed into a youth-friendly internet-enabled HIV risk calculator. A few participants (two women and three men) did not return for the second PGD. Participants were first reminded of the intention to develop a youth-friendly internet-enabled HIV risk calculator and were shown an early-stage prototype (Fig 2)—comprising five questions relating to age, gender, race, HIV status, and partner's HIV status—via an online link on a tablet. Finally, participants discussed the first five questions in a group setting to assess the relevance and comprehensibility of the questions on the HIV Risk calculator.

During the second PGD participants were asked to explore their preferences for mobile phone-based Apps—ways in which an internet-enabled HIV risk calculator could assess HIV risk and increase awareness of HIV risk among young people, potential barriers, and facilitators to using it, and what additional information should be included in the HIV risk calculator. See interview guide for PGDs (S1 File), for questions participants were asked in the group discussions.

**In-depth interviews.** Following the PGDs, 18 IDIs with young people who did not participate in any of the PGDs were conducted by trained female interviewers using a semi-structured discussion guide. IDIs were conducted in the participants' preferred languages, which were IsiZulu, Sesotho and English. Before the discussion, IDI participants completed a paper-based self-administered, short five-minute socio-demographic questionnaire, a questionnaire on mobile phone usage, and a 10-minute paper-based HIV and STI risk assessment questionnaire. The IDIs covered topics on HIV risk perceptions, sexuality, and possible challenges specific to young key populations and their experiences when wanting to access HIV testing services. The interview guide included sensitive questions, specifically directed to young key population groups. During the interview, participants were shown the HIV risk calculator prototype (Fig 2) on a tablet and completed the five questions with the interviewer. IDIs concluded with a discussion around the aesthetics and design of the HIV risk calculator. See interview guide for IDIs (S2 File), for details on the type of questions asked.

**Group discussion with ACAB.** Following the completion of the IDIs and PGDs, feedback was sent to the software developers to implement the participants' suggestions arising from the PGDs and IDIs. A group discussion with members of ACAB—comprising a mixed group of five young people aged 18–24 years (two males and three females)—was then conducted to test

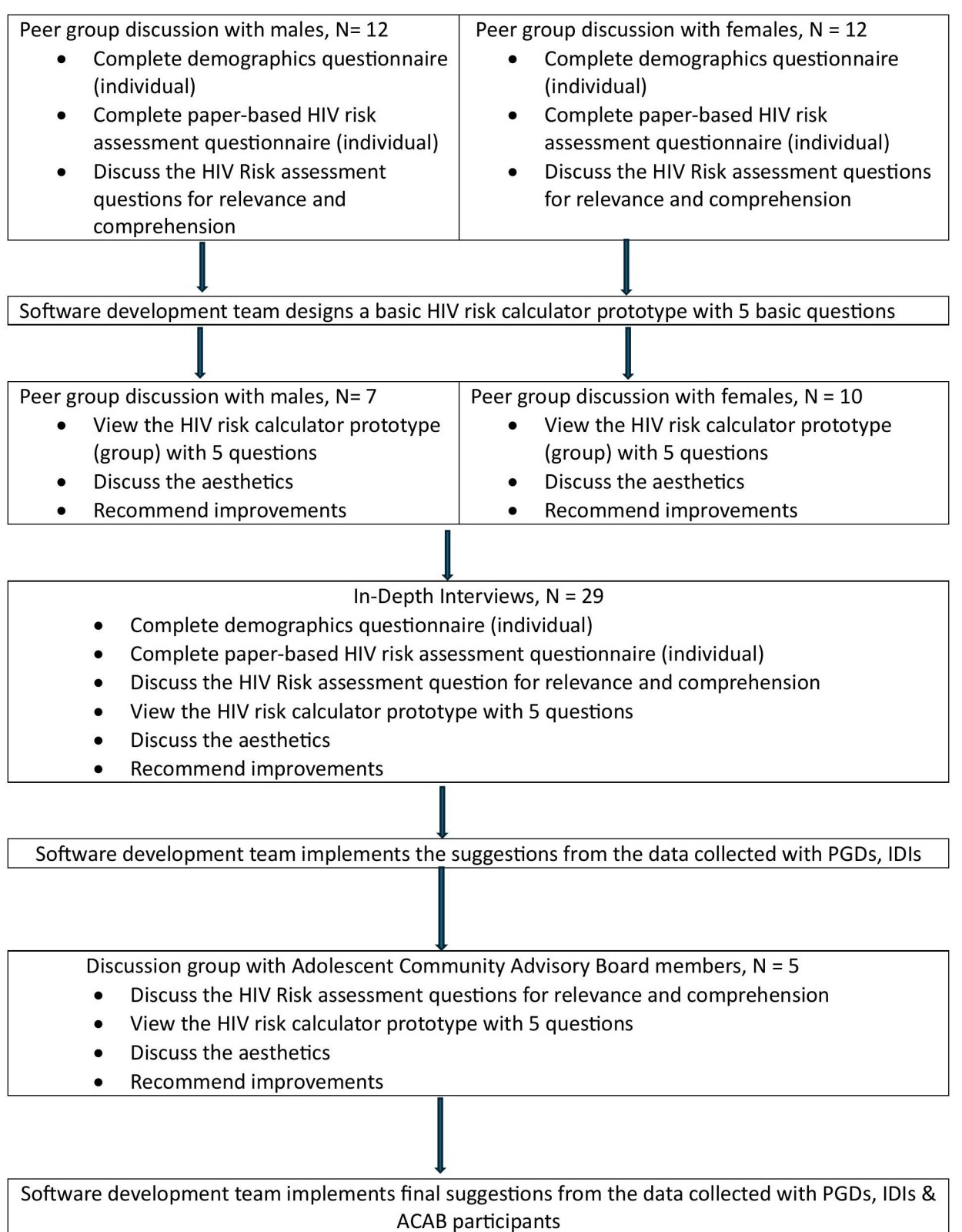

**Fig 1. Representation of study data collection process.**

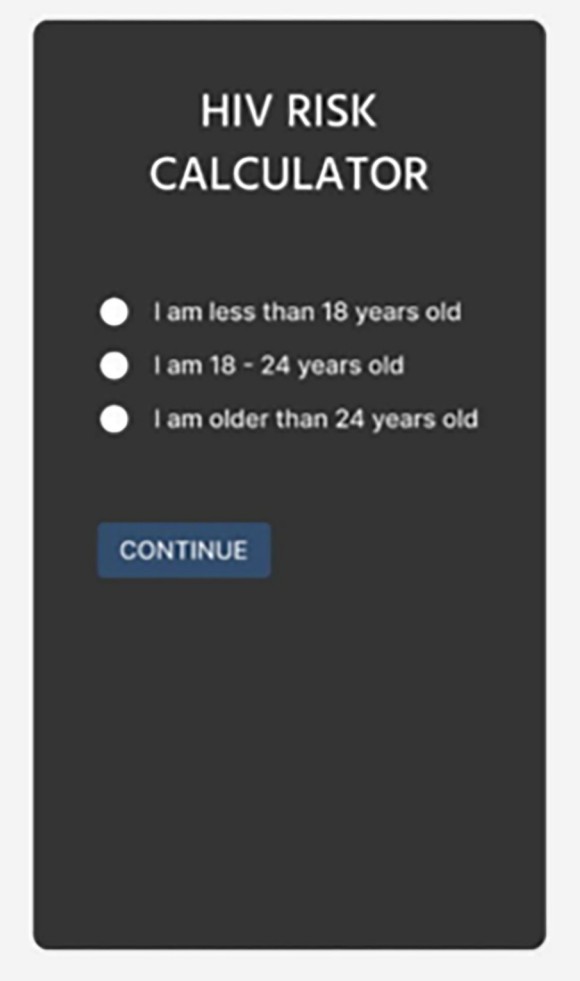

**Fig 2. HIV risk calculator first prototype.**

the prototype and solicit opinions on how the HIV risk calculator can be further improved for HIV risk assessments with young people. Participants were shown the HIV risk calculator prototype, asked to interact with it and then provided feedback.

**Socio-demographic questionnaire.**   The demographic component of the questionnaire comprised questions about age, gender, marital status, level of education, previous experience testing for HIV and undergoing HIV risk reduction counseling.

**Mobile phone usage questionnaire.**   The section on mobile phone usage covered questions on mobile phone ownership, personal use, and access to a mobile phone and the internet.

**HIV and STI risk assessment questionnaire.**   This questionnaire was derived from the National HIV Counselling and Testing guidelines developed by the National Institute for Communicable Diseases [31]. HIV and sexual risk behavior questions on the HIV and STI risk assessment questionnaire related to type, number and age of sexual partners; frequency of vaginal, oral and anal sex; sex-related condom use; sex under the influence of substances (i.e., alcohol and drugs); sharing needles when injecting drugs; tattooing and piercing behavior; and emotional health. The type of sexual partner was defined as: husband/wife (referring to a

sexual partner to whom the participant was married); dating (a sexual partner who was the main or regular partner, even if not living together); living together (a sexual partner with whom the participant was living, even if not married); or one-night stand (a sexual partner with whom the participant has had sex with for the first and one time only). To assess the history of STIs other than HIV, the questionnaire asked the following question: "have you ever tested positive for STIs such as gonorrhea, syphilis, or herpes?" The HIV and STI risk assessment questionnaire also included a question about emotional health: "Have you ever suffered from depression/stress?"

**Semi-structured interview guides.** Semi-structured interview guides were used to facilitate the PGDs and IDIs and covered topics relating to HIV testing experience, barriers and facilitators, and to evaluate the questions on the HIV and STI risk assessment questionnaire. The guides were also used to explore perceptions about an internet-enabled HIV risk calculator and ways in which the HIV risk calculator could assess HIV risk and increase awareness of HIV risk among young people. The semi-structured interview guide for IDIs included specific questions for young people who identified as LGBT individuals, MSM, and female sex workers. The following additional questions were included in the IDI guide: "Given the nature of your work (as a sex worker), what challenges do you think you still experience in accessing HIV testing services?" (only asked of sex workers); "Given the nature of your sexual orientation, what challenges do you think you still experience in accessing HIV testing services?" (asked of LGBT individuals and MSM only); "How can the HIV risk calculator be most effectively used through mobile phones and internet access?" (asked of LGBT individuals, MSM and sex workers only).

**HIV risk calculator.** The youth-friendly internet-enabled HIV risk calculator is an internet-enabled webpage with 30 self-reported item questions about HIV risk behavior. The webpage was developed in partnership with Dekode, a software development company in South Africa led by a young medical doctor and entrepreneur in digital technology and mHealth app development. The process of developing the HIV risk calculator began with building in the HIV risk assessment questions which were presented to participants for review during the PGDs and IDIs. We then conducted a thematic analysis of data based on PGDs and IDIs to identify preferences and suggestions for building a youth-friendly internet-enabled HIV risk calculator. The aim of the HIV risk calculator is to provide an opportunity for young people to assess their own risk for HIV infections by anonymously answering a series of questions about sexual practices, preferences and possible exposure to HIV infection. Based on the responses given by the user, the HIV risk calculator will generate a report with information about the possible type of risk identified and what the user can do to reduce their risk for HIV infection. The HIV risk calculator is self-administered and anonymous.

## Data analysis

Qualitative data were analyzed manually using a framework analysis approach [32]. First, a list of codes was extracted from the PGD and IDI interview guides to obtain the initial framework of codes on a Microsoft Office Excel spreadsheet. The first two PGD and two IDI transcripts were read and coded by the lead researcher (first author) and an MA student (second author) based on the initial framework, while allowing identification of emerging codes. The list of codes was entered into NVIVO software for qualitative analysis and allowed for coding the remaining transcripts. The remaining transcripts were coded by the lead researcher following a line-by-line technique to capture codes that would otherwise emerge from data that was not initially part of the pre-existing codes. This process was followed until a comprehensive codebook was developed. A list of final codes was presented to the research team involved in this

study. Final codes were discussed and approved by the supervising researchers. Codes were reviewed to form themes and sub-themes, which involved categorizing similar and diverging codes. Data was analyzed based on two key ideas: evaluating the HIV risk assessment questionnaire and developing a youth-friendly internet-enabled HIV risk calculator. Socio-demographic characteristics and mobile phone usage data were described using SPSS Version 25.0 [33].

### Reflexivity

As part of reflexivity, the researcher MM shared her goals and her interest in the research on HIV prevention for young people. As the researcher, MM was a PhD student during data collection and has had extensive research experience in qualitative data collection and analysis. The research assistants involved in data collection were both MA students and received training in qualitative research methods. The researcher and research assistants had no prior knowledge or interaction with the participants. A rapport was built through initial introduction to the participants prior to the PGDs and in-depth interviews. Female participants were interviewed by the female research assistant and the male participants were matched with the male interviewer, whenever possible.

MM's initial assumptions were that young people are not aware of their risk for HIV and other sexually transmitted infections. Through the discussion with the participants, it became a reality that young people cannot define what HIV risk is and did not understand the importance of completing an HIV risk assessment questionnaire during HIV counseling and testing. This realization may have biased the data analysis; hence, the data analysis involved two researchers and went through the research team for approval of codes and themes generated during the analysis.

### Ethical considerations

This study was approved by the University of the Witwatersrand Human Research Ethics Committee (HREC) (non-medical, ethics reference number H14/09/04). All participants provided written informed consent for completing the HIV and STI risk assessment questionnaire and for participating in a PGD or IDI. Prior to commencing any study procedures, participants signed an informed consent form for the study and a separate informed consent form for the audio-recording. Participants were reimbursed R150 ($8.25) to cover the costs of transport and refreshments.

## Results

### Participant demographics and mobile phone usage

Table 1 provides a detailed description of the demographics, illustrating forty participants (23 cisgender males, 15 cisgender females, and 2 transwomen), participated in either a PGD or IDI. The median age was 20 (interquartile range 19–22.75), and 58% (23/40) were male. Of the 40 participants, 30 self-identified as straight, 9 as lesbian or gay, and 1 as bisexual. Two participants identified as female sex workers. Seventy eight percent (78%) self-reported HIV negative status and 13% were not sure of their HIV status. Mobile phone ownership was 88% (35/40), of which 97% (34/35) used the prepaid method for accessing mobile airtime, data, or SMS bundles. Overall, 72.5% (29/40) had access to the internet via a mobile phone, tablet, laptop, or computer.

### Feedback on the risk assessment for HIV and STI questionnaire

PGD and IDI participants were asked to identify HIV and STI risk assessment questions that they found difficult to comprehend and were asked to suggest how the questions could be modified or clarified to a level of language and wording that young people would understand.

**Table 1. Participant demographics for IDIs and PGDs.**

| Variables | Values | N (%) = 40 |
|---|---|---|
| Age | Median age | 20 (IQR-19-22.77) |
| Gender | Cisgender male | 23 (57.5%) |
|  | Cisgender female | 15 (37.5%) |
|  | Transgender | 2 (5%) |
| Race | Black | 40 (100%) |
| Sexual Orientation | Heterosexual | 30 (75%) |
|  | Lesbian/Gay | 9 (22.5%) |
|  | Bisexual | 2 (2.5%) |
| Highest level of education | Completed grade 12 | 23 (57.5%) |
|  | Less than grade 12 | 12(30%) |
|  | Completed primary | 1 (2.5%) |
|  | Missing | 2 (5%) |
| Employment status | No | 38 (95%) |
|  | Yes | 2 (5%) |
| Phone ownership | Yes | 35 (88.0%) |
|  | No | 5 (12%) |
| *Pay Airtime | Prepaid | 35 (100%) |
|  | Contract | 0 (0%) |
| Internet | Yes | 29 (72.5%) |
|  | No | 7 (17.5%) |
|  | Missing | 4 (10%) |
| HIV status | Negative | 31 (77.5%) |
|  | Positive | 1 (2.5%) |
|  | Not sure | 5 (12.5%) |
|  | Missing | 3 (7.5%) |

\* Pay Airtime is taken from SPSS variable names derived from the question: "how do you pay for airtime?"

Four categories of feedback were identified based on discussions with PGD and IDI participants: (1) confusing questions, (2) difficult words in questions, (3) vague questions, and (4) controversial questions. Table 2 presents the list of questions identified as problematic and indicates how these were revised following the PGDs and IDIs.

*"They [the questions] are straight out clear and it's simple English. . .but then I think the questions on this App, they need to be optional whether in English or in Zulu or in whatever language that they may prefer."* **(IDI-013, transgender, 20 years old)**

*"[The] English is straight forward. There's no school that doesnt teach English. Even if I do English at a certain age, this and that. No one must complain about that English. After all, we are not in primary."* **(PGD, females, 18–24 years old)**

However, in some questions, participants identified words they considered unfamiliar to them, which needed to be revised or removed. Other participants suggested that the HIV risk calculator be made available in all South Africa's 11 official languages, to cater to young people for whom English is not their primary language.

*"Like, before you can answer questions, you must choose which language do you wanna use. Maybe if IsiZulu, then questions must be in [Isi]Zulu. Sometimes, isiZulu does make sense."* **(PGD, females, 18–24 years old)**

**Table 2.** List of HIV and STI risk assessment questions identified as problematic, representative extracts, and revised version of the questions.

| Type of feedback received on questions | Original question from HIV and STI risk assessment questionnaire | Quote extracts from participants | Revised questions from the HIV and STI risk assessment questionnaire following participant feedback |
|---|---|---|---|
| (1) Confusing questions | Which of the following sexual activities do you engage in? Tick all that apply:<br>□ Receive oral sex without condom/Receive oral sex with condom<br>□ Receive anal sex without condom/Receive anal with condom<br>□ Receive vaginal sex without condom/ Receive vaginal sex with condom<br>□ Receive oral-anal sex without condom/Receive oral-anal sex with condom<br>□ Give oral-anal sex without condom/Give oral-anal sex with condom | *"Yeah, there are some questions there on sexual behavior, I know like some of us we didn't know. We didn't like those questions—receiving anal or oral sex without a condom. Cause these questions, what can I say? It is too complicated. I didn't understand the question."* **(PGD males, 18–24 years old)** | What type of sex do you have? Tick all that apply:<br>□ Oral sex without condom/Oral sex with condom<br>□ Anal sex without condom/Anal sex with condom<br>□ Vaginal sex without condom/Vaginal sex with condom |
| | | *"Is this one about oral and anal . . . to receive sex without a condom, how do I receive myself, because I am a guy?"* **(IDI-001 male, 18 years old)** | |
| | | *"I was confused. I only answered 'Received oral sex without a condom'. Then I was like, nope, 'Give oral sex without a condom'."* **(IDI-006 male, 24 years old)** | |
| | | Participant: *Ok maybe I have–problem with this, the language. There are terms that I didn't understand. Like when you [are] talking about anal, oral . . . like virginal [vaginal] sex, I didn't know.*<br>Interviewer: *You just know sex?*<br>Participant: *Yeah, sex and virgin.* **(IDI-005 male, 19 years old)** | |
| (2) Difficult wording in questions | Have you engaged in group sex (**orgy**)? | *"I didn't understand the question . . . I didn't know the meaning of orgy. It's the first time [I am] seeing it."* **(IDI-001 male, 18 years old)** | Have you engaged in group sex? |
| | How often do you have **penetrative** vaginal/anal sex? | *". . . and then the penetrative, I didn't know the penetrative word what it meant but now I do. So, I think some of the questions, some of the words here they would at least change them or maybe in brackets add a simpler term of the word."* **(IDI-016, female, 22 years old)** | How often do you have vaginal/anal sex? |
| (3) Vague questions | Have you tested for HIV? | *"No, I just have this question . . . Have you tested for HIV? As in when? 3 months back, in your lifetime or when? It should be like in the last 3 months. Have you ever tested? In the last decade? There is a difference from 10 years and now, probably in those 20 years, you had multiple or engaged in sex with a positive partner."* **(PGD-ACAB mixed gender, 18–24 years old)** | When was the last time you tested for HIV? |
| | Do you drink alcohol? How often? | *"I think it will be better if you ask, 'Does alcohol work for you in your life?' than asking 'How often do you drink?' Because no one would want to say that they are drunkards. I'm not saying this fit everyone; there is someone who is going to say 'yes', and they drink once a week. But when the question is like this, it pushes me towards saying 'no'."* **(PGD males, 18–24 years)** | Have you had any type of sex when you have been drinking alcohol or doing drugs? |
| (4) Controversial questions | What is your race? What is your partner's race? | *"The question about race. . . it's either a false result or something. Like it doesn't make sense . . . because if I fill in all this form then change from Black and say I am Indian, if it gives me a different result that means it is being racist. That question is unnecessary because I don't believe it give us [the]exact answer."* **(PGD males, 18–24 years old)** | This question was not removed from the HIV risk assessment question list, as it was misconceived by the young people. However, the reasoning behind the question was explained to them. |
| | | *"I believe that maybe people think to themselves that HIV is mostly common with Black people, so if I have sex with a white person, then I won't get infected."* **(PGD females, 18–24 years)** | |
| | | *"Already we know that . . . I'm not sure if it's proven but generally, we know that HIV it's mostly associated with Black people. But it's not ok, like, to put this race thing as part of the questionnaires. Cos once you say Black, it increases the chances of you being infected. I feel that it should be taken out."* **(IDI-004 male, 20 years old)** | |

Overall, participants understood most questions on the risk assessment questionnaire. The HIV and STI risk questions were in English, and most agreed that the level of language used was relatively easy to understand.

*"Ja ja, translations as well they are also important because inclusivity is crucial"* **(IDI-011, female, 24 years old)**

Certain questions were identified as confusing and difficult for young people to understand, while others sparked debate and were questioned for their relevance or appropriateness in assessing HIV risk. For example, a few IDI and PGD participants perceived the question regarding race (i.e., What is your race and what is your partner's race?) to be irrelevant to HIV risk. The question raised insightful debates across the PGDs and among some IDI participants. Most did not like the question, stating that asking about race prejudices Black people as the population that is HIV infected more than other racial groups.

*"If you tick every question and then you change the race question, maybe you choose black, I mean Indian and then the percentage changes, yeah that's offensive. It means the question must not be asked."* **(PGD males, 18–24 years old)**

*"Yes, why? We live in a democratic society. Do you have to tell me, ask about my partner's race? I think that's irrelevant as well."* **(IDI-010, male 24 years old)**

While some participants felt that the question is offensive, other participants misconstrued the relevance of the race question in the questionnaire. For example, a participant in a peer group discussion with males, suggested that the race question is for statistical purposes to prove that HIV came from white people and gay men.

*"[the question about] race is very clear. As the gent said earlier, HIV was brought by white people and gays so by asking race you will be able to find the average number even though the painful part is that we will not get the exact number from the U.S, but we will be able to get our own number. We are not going to have sufficient evidence, but it will help to know that as black people how many are affected by HIV, do you get me? We will compare the number of blacks, coloureds, and Indians and if we win it will prove that they came with HIV."* (PGD males, 18–24 years old)

Another participant in the PGD with females, suggested that the race question is to show that most black people from the village are affected by HIV because they love sex and lack of access to condoms when compared to white people.

*"I think it [race question]is very important because as people we are different and most people who live in the villages are not exposed to things such as condoms so race can clarify that in this province it is mostly black people, they live in the village and they don't have facilities to access condoms to protect themselves or maybe to see if black people love sex more than white people you see."* **(PGD females, 18–24 years old)**

### Feedback on developing a youth-friendly internet-enabled HIV risk calculator

Four themes were identified from the discussion with PGD and IDI participants on developing a youth-friendly internet-enabled HIV risk calculator: (1) Preference for an app vs. a webpage, (2) privacy and confidentiality, (3) colorful, fun, and discrete interface, (4) an interactive HIV and STI risk assessment and information providing tool. Table 3 provides an overview of participants' suggestions and preferences for a youth friendly HIV risk calculator and how these were used to guide the design process and adaptations of the HIV risk calculator.

**Table 3. Participant preferences for a youth-friendly HIV risk calculator and representative extracts.**

| Preferences for a youth-friendly HIV risk calculator | Extracts from participants | Solutions for the HIV risk calculator |
|---|---|---|
| **App vs. webpage** | *"I think that the app is the best option since we are youth and mostly everything we do is apps, even the photo edits we use apps. We are used to using apps, so it will be really easy for us to use. Young people have apps on their phones, so with the website we will complain about data."* (**PGD ACAB, mixed gender, 18–24 years old**) | • Based on participants' suggestions to have an interactive, youth-friendly app, we developed a chatbot solution.<br>• The chatbot provides a real-time interactive experience. |
| | *"It should be in an app because a webpage—okay does this take much data? [Be]cause people fight a lot when it comes to data, so I think it should be more of like maybe an app. Maybe there should be a data for this specific app, like maybe WhatsApp. So, I think there should be an app rather than a website."* (**IDI-016-female, 22 years**) | |
| | *"App is better. Whenever you wanna do something or check something, it's there. A webpage—it's like a long process to get there and everyone is app crazy now."* (**IDI-011-female, 24 years**) | |
| **Privacy and confidentiality** | *"You see the internet stuff, like the ones that are connected to the internet, I don't trust them. So, if I was given like a guarantee that if you put your information here, nobody else will open it. Maybe if you want to answer that thing [HIV risk calculator] and after you provided a password and you lock it down, ah that way, I will be honest 100% because I know that this will be known by me and that person [researcher]."* (**IDI-008, male, 23 years old**) | As part of the **HIV risk calculator** development, emphasis was placed on ensuring that no identifying information, such as names and date of birth were captured, and that the interaction is kept anonymous.<br>• The chatbot does not retain participants' questions and responses once the window is closed. |
| | *"I won't be able to tick my HIV status and that of my partner. HIV results are private and is not something you go around writing everywhere"* (**PGD, males, 18–24 years old**) | |
| | *"As soon as I leave the app, the data should be gone. It should be written off for in case someone opens the app and then it's like a clean sheet for someone to calculate"* (**PGD ACAB, mixed gender, 18–24 years old**) | |
| **Discrete interface** | *"If it is written **HIV risk calculator**, if my mom sees that, she will be curious and wanna know more and then I'll also be putting myself in danger in a way. Because by then she will know that I'm sexually active and what's going on"* (**PGD ACAB, mixed gender, 18–24 years old**) | • Participants agreed that the app design and logo must not depict anything related to HIV.<br>Participants emphasized that they prefer an HIV risk calculator that provides HIV-related information discretely to maintain user confidentiality and protect young people from having other people, particularly parents, find out that they are using the app.<br>• Therefore, the logo includes the letter 'q' for 'Quick Questions', which does not have the appearance of being related to HIV. |
| | *"Yes, when it [**HIV risk calculator**] appears on the home display, it shouldn't have anything indicating what the app is about; it shouldn't be obvious what the app is about. Only the user and the owner of the phone should know what the app is about."* (**PGD ACAB, mixed gender, 18–24 years old**) | |
| | *"The reason we wouldn't want to have something that is more explicit if it's an app. I'm under my parents' guidance so if they were to find such an app on my phone, already that's alarming to them and sometimes you might find that I'm not really sexually engaged but I'm just curious and interested about finding out more on sexual infections."* (**PGD ACAB, mixed gender, 18–24 years old**) | |
| **Colorful interface** | *"It needs to be colourful and it needs to be appealing and then again it must have that vision man!"* (**IDI-013, transgender, 20 years old**) | • Having a colorful background or layout was raised as a key feature of the interface, which would make the app appealing and attractive to young people.<br>• Having pictures and visuals that are relevant for young people and that they can identify with, would also make it more attractive, so that young users are not easily bored.<br>• The interface includes different personas that young people could choose from based on their own preference (i.e., option slay vs. afrochic vs. finesse). |
| | *"I think it could look more colorful. I think it could benefit from having a bit of animation, motion animation and then a few sparkles man, yes just to make it more user-friendly for the youth."* (**IDI-013, transgender, 20 years old**) | |
| | *"It must at least tell people that this thing does not mean you have HIV or not, it's a risk calculator . . . it must tell you that to avoid being at risk, okay condomise, decrease number of partners, you see? It must also tell you . . . to get tested, go to your nearest clinic and stuff like that."* (**PGD, females, 18–24 years old**) | |

*(Continued)*

**Table 3.** (Continued)

| Preferences for a youth-friendly HIV risk calculator | Extracts from participants | Solutions for the HIV risk calculator |
|---|---|---|
| **Interactive HIV risk assessment and information giving tool** | *"Like they should put in a summary of the stuff you should do. Like you should do this and that in order to lower your risk of actually being exposed to the virus."* **(IDI-016, female 22 years old June 2018)** | • The interactive aspect should allow young people to learn more about their HIV risk, health related information and accessing HIV testing services.<br>• As part of the design once a young person has completed the HIV risk questions, an HIV risk profile is generated tailored towards participants responses.<br>• A feature was added that provides information to test at PHRU's youth-friendly HST service at no cost. |
| | *"It is possible that in this app they add a doctor (so that you can ask questions)? Maybe you don't feel satisfied by the questions."* **(PGD, females, 18–24 years old)** | |
| | *"If it's an app you get notifications to say I think you should assess your risk again if it's been six months."* **(PGD ACAB, mixed gender, 18–24 years old)** | |

## Preference for an App vs. a webpage

Participants in this study preferred a mobile App HIV risk calculator to a website because the mobile App version will be quick and easy to access and cost less data usage.

*". . .besides the webpage being more expensive than the app. Umm I think the app you can access it very easy and quicker than the webpage cause with the webpage you probably have to go to your browser, after your browser you click www dot blah blah dot co dot ja whatever, unlike the app it's one click away"* **(PGD ACAB, mixed gender, 18–24 years old)**

## Discrete interface

Participants advised that the HIV risk calculator should not be obvious that it is related to HIV. They suggested a discrete interface especially if it is a mobile App version. Young people preferred discrete and private means of assessing their own risk for HIV. While participants were eager to receive notifications or reminders from the online HIV risk assessment to test for HIV every three months, the majority concurred that the design and logo of the internet-enabled HIV risk assessment should not contain any HIV-related imagery.

*"If it's in a web[page]. . .a website it should say a HIV Risk Calculator then in [an]App it should be something different then [only]if you have access to it you will see what is happening [what the App is about]"* **(PGD ACAB, mixed gender, 18–24 years old)**

## Privacy and confidentiality

Participants emphasized the need for privacy and confidentiality of the HIV risk calculator. Some participants shared their fears about being found having an App for HIV and subsequently being perceived to be sexually active. During an in-depth interview one participant expressed their need for confidentiality of the information about sexual risk behaviours shared on the HIV risk calculator.

*"If I'm in a relationship with more than one partner, I wouldn't want my partner to know about my sexual risk behaviour. I wouldn't want that person to have access to such that I have 3 partners"* **(IDI-006, Male, 18–24 years old)**

## Colorful interface

Participant stated that the App should be eye catching, colorful and have graphics to grab the interest of young people.

> *"I'm a person who loves color so they should add a little bit of [color]. It should look a bit like a game so, like, people can at least be attracted to it, get to try it."* **(IDI-013, transgender, 20 years old)**

The same participant suggested that researchers and developers should avoid "being boring" and learn from other social media apps that are appealing.

> *"Just try not to make something so monotone. The reason why the Instagram(s), the twitter(s) and everything take on so wonderfully is that it's very graphic, you know".***(IDI-013, transgender, 20 years old)**

Participants reiterated their preference for an interactive and informational online-based HIV risk assessment, which would allow them to ask questions openly about their health and identify nearby youth-friendly clinics for HIV testing.

Throughout the software development phase, participants' opinions, and recommendations from PGDs and IDIs were considered in developing an HIV risk calculator prototype incorporating a chatbot platform. Fig 3 illustrates the logo and personas, while Fig 4 illustrates the final HIV risk calculator prototype elements, and Fig 5 illustrates the final prototype front page.

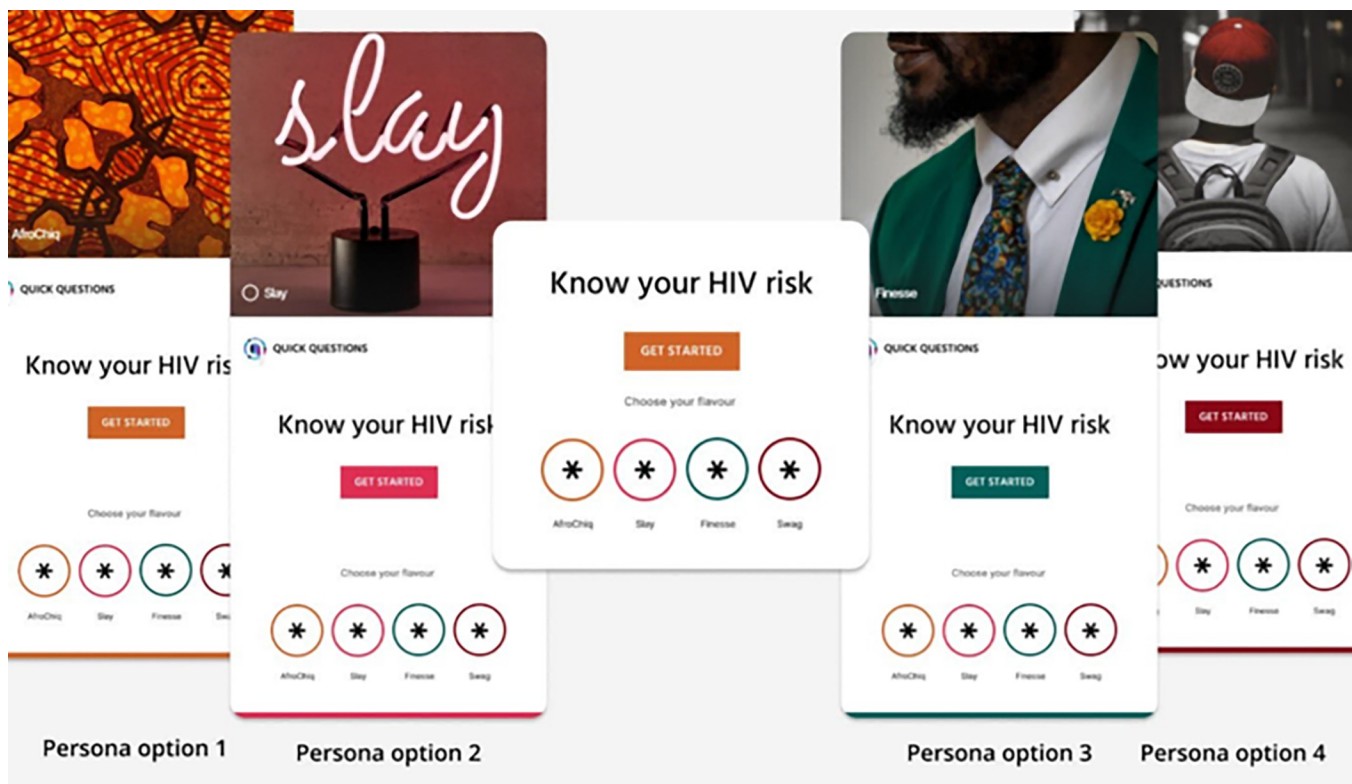

**Fig 3. HIV risk calculator logo & personas.**

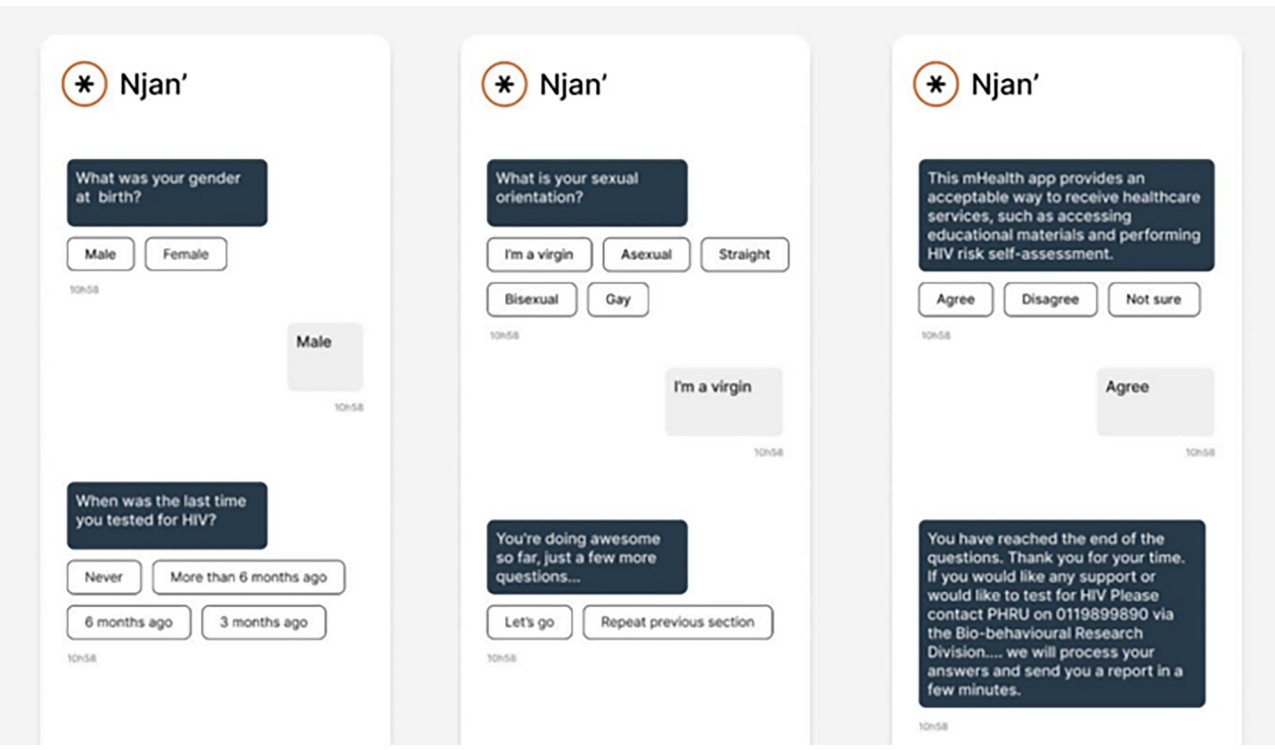

**Fig 4. HIV risk calculator final prototype with example questions.**

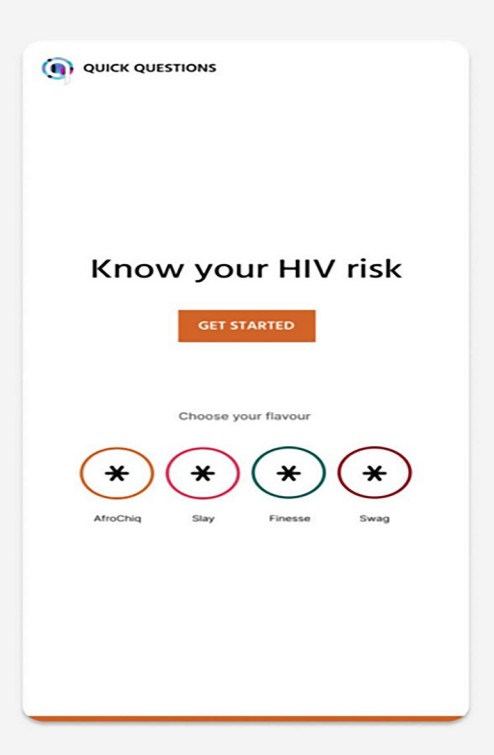

**Fig 5. HIV risk calculator final prototype front page.**

## Discussion

Our study is one of only a few South African studies to take a participatory approach to developing an HIV risk calculator for young people. Using peer group discussions and in-depth interviews, young people in collaboration with researchers were able to develop a youth-friendly internet-enabled HIV risk calculator for self-assessment of HIV risk. A youth-friendly HIV risk calculator suggests an age and context appropriate, accessible, discrete, and confidential HIV risk tool. Our study applied the Principles for Digital Development framework [34], focusing on four of the nine principles. The nine principles being: *design with the user; understand the existing ecosystem; design for scale; build for sustainability; be data driven; use open standards*, *open data*, *open source and open innovation; reuse and improve; address privacy and security and be collaborative* [34]. By applying the four principles of digital development, being: *design with the user; reuse and improve; address privacy and security and be collaborative*, our study was able to collaboratively develop an HIV risk calculator that is youth-friendly, acceptable, confidential, discreet and user interface design.

Our data revealed the value of obtaining young people's input when developing a mHealth tool. This is similar to other studies, some of which were in South Africa, that used participatory research to co-develop mobile health applications for HIV prevention among different populations including young people, adolescent girls and young women as well as MSM [18,35,36,37,38]. These studies revealed the importance of including target populations in the development of mHealth for HIV prevention. In their study in Peru, which was aimed to develop an SMS text message intervention on sexual and reproductive health with adolescents and youth, Guerrero et al. [18] explain how they had to rephrase certain messages to the level of understanding of adolescents and youth in Peru. For example, study participants suggested the use of words that youth prefer, such as "boyfriend/girlfriend" instead of "intimate partner" and "v zone" instead of "vagina" to grab the attention of adolescents and youth. Similarly, in this study, we identified certain questions in the HIV risk assessment questionnaire that were not at the desired level of understanding for young people in a peri-urban South African setting. Guerrero et al. [18], revealed the importance of involving adolescents and young people in co-creating age and culturally appropriate mHealth applications. All questions considered problematic were revised in collaboration with participants and discussed further with the research team for corroboration. Questions approved by participants and the research team were then sent to the software developer to incorporate into the HIV risk calculator.

Findings from this study also show that the HIV risk calculator, as a confidential and anonymised self-assessment for HIV risk, can potentially reduce under reporting or over reporting of HIV risk practices. Similar to our findings, Dietrich et al. [35], revealed that participants identified confidentiality and privacy as key factors that can enhance participants' confidence and comfortability in answering intimate questions. Thus, this could potentially reduce the challenge of providing socially desirable responses inherent in HIV risk assessments conducted in clinics [18,35].

Findings from the PGDs and IDIs for the HIV risk assessment questionnaire showed that the participants understood most of the questions. The main areas of concern were the phrasing and relevance of the question [36].

Additionally, the HIV and STI questionnaire lacked definitions of terms relating to sexual activity, which was a concern, as these are crucial for recognizing potential risks. Therefore, refining some questions was essential to ensuring that young people understood and responded appropriately to the HIV risk questions. Most concerning was the finding that revealed controversy surrounding the question about race in the HIV risk assessment. Some young people did not understand why the question about race is included in the HIV risk

assessment questionnaire. Further concern was to learn about the misconstrued perception that young people in the study held about that question on race. In our knowledge, there are no studies that explore young people's understanding of the questions included in the HIV and STI risk assessment questionnaire, particularly in South Africa. Therefore, we can confirm that allowing young people to scrutinize the questions posed for assessing HIV risk is a critical step in ensuring that questions are age-appropriate and context-relevant. This study emphasizes the need for a youth-engagement approach to developing HIV prevention interventions that are acceptable and effective for young people. While several mobile apps have been developed for HIV prevention [20,21,22,23], there is limited evidence of the extent to which they are being used and are considered acceptable by the target audience. This study adds to the limited existing literature on mobile phones on HIV risk assessment, and behavioral data collection among young people in South Africa [15].

The responses from the PGDs and IDIs provided valuable insights into the acceptability of the HIV risk assessment questions and guidance on the design of the HIV risk calculator.

Designing a reusable, yet simple app that accurately reflects both current HIV risk and change in risk over time entails balancing the app's efficacy with the responsibility of informing users of how each criterion is used. For instance, using certain sensitive criteria such as race or ill-defined questions can mislead users into reaching erroneous conclusions about the role these factors play in risk outcomes. Comparing different risk measurements yielded by different answer choice combinations can give rise to problematic assumptions that correlative criteria cause HIV exposure, which may reinforce stigma and assumptions regarding the HIV patient demographic. This resembles the findings of a literature review that revealed that strategies employed in the communication of HIV risks had the potential to cause unintended adverse effects, including stigmatization [37]. Certain participants voiced concerns about questions that did not correspond with their sexual identity or beliefs about the range of what they deemed to be acceptable sex actions. Thus, to avoid ambiguity or skewing data with preconceived biases, it is crucial to clarify subjective nuances in criteria with specific technical details such as time, frequency, and physical description. And it is crucial for HIV prevention organizations to ask themselves continually whether young people understand their messaging.

Findings from the PGDs and IDIs also indicated that financial accessibility, privacy, and interface design are vital to creating an effective HIV risk calculator. Data availability and costs are vital factors in marketing mHealth apps, particularly in low-and middle-income countries. Our study participants preferred an app platform over a website because of data constraints. However, some participants voiced concerns that even an app might not be the most equitable platform, given that smartphone access varies across different socio-economic backgrounds. This highlights the importance of creating a data-friendly app that places minimal financial constraints on users. In South Africa, the government has taken the initiative to give young people internet access for educational purposes. While initiatives such as Project Isiswe have made Wi-Fi available in several areas in Soweto [38], there is a need to ensure our app is equitable and accessible to the most vulnerable and disadvantaged populations if we are to succeed in our efforts to provide an effective tool in the prevention of HIV in the youth of South Africa.

Social stigma is one of the biggest barriers to HIV prevention in youth [39,40,41]. As with other studies on HIV prevention using mHealth, [42,43], the participants in this study stated that they preferred an app that prioritised user confidentiality.

When creating the HIV risk calculator app, we discovered young people's preference for an interactive and colorful interface. Designing a user-friendly interface was critical to maintaining adherence. Considering the information fatigue already reported in several studies on HIV prevention, creating an interactive app presented an excellent opportunity to provide young

people with information they want or choose to receive [44,45]. The modifications made to the HIV risk assessment questions and the HIV risk calculator through the collaborative process may present the opportunity for improved accuracy of the HIV risk assessment and acceptability of the App among young people in South Africa.

## Limitations

The collaborative nature of the PGDs and IDIs meant that participants may have responded more favorably than if the process had been less participative. Because only young people who were approached and available could take part in the study, the findings are not generalizable to all young people in Soweto, South Africa. The study could be improved by using surveys to rate the HIV risk calculator's usability, usefulness and acceptability. While participants could identify items that were difficult or vague, some of them did not understand the meaning of HIV risk, and therefore could not suggest how to improve certain question items. Fluency in English was not a criterion for participation, as the study did not want to exclude young people who were not in school. IDIs, PGDs and group discussions were conducted in the participants' preferred languages. However, the demographic questionnaire, the HIV and STI risk assessment questionnaire, and the HIV risk calculator questions were in English and not translated into other local languages spoken in Soweto (IsiZulu, Sesotho, IsiXhosa), which may have influenced participants' responses. The study was a small-scale research project and therefore could not apply some of the principles outlined in the Principles for Digital Development which included: understanding the existing ecosystem; design for scale; build for sustainability; be data driven; use open standards, open data, open source and open innovation; reuse and improve. This was mainly due to the limited scope of the research project and therefore further studies can be conducted to improve the uptake, accessibility of the HIV risk calculator and influence policy in HIV prevention efforts in South Africa.

## Conclusion

The research team discovered the value of taking a collaborative approach to developing an mHealth app for assessing HIV risk and aiding prevention. We found that privacy, confidentiality and ease of use are crucial to promoting acceptability and a willingness to use novel and internet-enabled HIV prevention tools.

Participants indicated a preference for an easy-to-use, interactive, real-time assessment that not only offers a discrete and private means to self-assess HIV risk, but that is also colorful, interactive and informational. In addition, to ensure effective communication, they suggested creating an app that has accessible language and avoids unnecessary literary barriers.

The young people in this study also proposed providing links to clinics or doctors for more information, which is encouraging and demonstrates their openness to accessing health care online.

## Declarative statements

Any opinions, findings, conclusions or recommendation expressed in this material is solely the responsibility of the authors and do not necessarily represent the official views of the funders.

## Supporting information

**S1 File. Interview guide for PGDs.**
(DOCX)

**S2 File. Interview guide for IDIs.**
(DOCX)

## Acknowledgments

The authors wish to thank the young people who took part in this study, and the adolescent community advisory board (ACAB) [name of clinic removed for confidentiality], which was integral to data collection. We also thank Dr Kathryn Hopkins and Prof Kennedy Otwombe for reviewing the first draft of the manuscript.

## Author Contributions

**Conceptualization:** Mamakiri Mulaudzi, Janan J. Dietrich.

**Formal analysis:** Mamakiri Mulaudzi.

**Funding acquisition:** Mamakiri Mulaudzi, Janan J. Dietrich.

**Investigation:** Mamakiri Mulaudzi.

**Methodology:** Mamakiri Mulaudzi, Gugulethu Tshabalala.

**Project administration:** Mamakiri Mulaudzi.

**Software:** Kofi Ebenezer Okyere-dede.

**Supervision:** Janan J. Dietrich.

**Writing – original draft:** Mamakiri Mulaudzi, Stefanie Hornschuh.

**Writing – review & editing:** Mamakiri Mulaudzi, Gugulethu Tshabalala, Stefanie Hornschuh, Kofi Ebenezer Okyere-dede, Minjue Wu, Oluwatobi Ifeloluwa Ariyo, Janan J. Dietrich.

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
