## [Decision Letter · Decision Letter 0]

9 May 2023

PDIG-D-23-00013

Developing a youth-friendly internet-enabled HIV risk calculator: A collaborative approach with young people aged 18–24 years living in Soweto, South Africa

PLOS Digital Health

Dear Dr. Mulaudzi,

Thank you for submitting your manuscript to PLOS Digital Health. After careful consideration, we feel that it has merit but does not fully meet PLOS Digital Health's publication criteria as it currently stands. Therefore, we invite you to submit a revised version of the manuscript that addresses the points raised during the review process.

Please submit your revised manuscript within 60 days Jul 08 2023 11:59PM. If you will need more time than this to complete your revisions, please reply to this message or contact the journal office at digitalhealth@plos.org. We look forward to receiving your revised manuscript.

Kind regards,

Jessica Keim-Malpass

Academic Editor

PLOS Digital Health

Journal Requirements:

1. We ask that a manuscript source file is provided at Revision. Please upload your manuscript file as a .doc, .docx, .rtf or .tex.

Additional Editor Comments (if provided):

Reviewers' comments:

Reviewer's Responses to Questions

**Comments to the Author**

1. Does this manuscript meet PLOS Digital Health’s publication criteria? Is the manuscript technically sound, and do the data support the conclusions? The manuscript must describe methodologically and ethically rigorous research with conclusions that are appropriately drawn based on the data presented.

Reviewer #1: Yes

Reviewer #2: Yes

2. Has the statistical analysis been performed appropriately and rigorously?

Reviewer #1: N/A

Reviewer #2: Yes

3. Have the authors made all data underlying the findings in their manuscript fully available (please refer to the Data Availability Statement at the start of the manuscript PDF file)?

Reviewer #1: No

Reviewer #2: Yes

4. Is the manuscript presented in an intelligible fashion and written in standard English?

Reviewer #1: Yes

Reviewer #2: Yes

5. Review Comments to the Author

Reviewer #1: Review – overarching comments 

Thank you for this paper, which describes inviting young people to pilot test and comment on an app to assess HIV risk prior to finalising the development of the app. Although an interesting read, the paper does not add sufficiently to the available evidence on co-development or mHealth; at times, the paper also lacks clarity on what the app would achieve (ie. How would accurate risk of assessment overcome barriers to accessing available services; at times the app seems to focus only HIV and other times all STIs) and why different populations of young people were asked to comment differently on the app. The discussion lacks critical reflection on how the findings relate to the available literature; and, although, young people were involved in designing the app, it’s not clear whether young people were involved prior to this – ie. In deciding that an app would be most likely to address their barriers to HIV prevention services.

It is unclear from the title, abstract and introduction that the study includes young key populations. Key populations are briefly mentioned in the abstract but then not in the introduction or under recruitment of PGD participants. It seems the focus was young people and young key populations – this should be made clearer in the title, abstract and introduction. 

Related to this, the PGD included young people and asked them to comment on the questions included in the risk assessment. The IDI focused on key population’s preference for an app vs internet-enabled risk assessment tool and its appearance. Why weren’t young people in general and young people that are members of a key population not asked both? This needs clarification, as features important to key populations may not be important to individuals who do not belong to a key population; similarly, key populations might have considered some questions to be missing, e.g. related to sexuality and gender expression to be lacking from the assessment tool. 

Later in the manuscript it states that the risk assessment is for HIV as well as other STIs, this should be made clearer in the introduction and provide information on STI burden (other than HIV) among young people in South Africa.

One limitation of the study is that, although the study involved youth in the development of an app, did the authors discuss with young people how best to address barriers to services? It would seem that an important first step would be to ask young people how to improve their access to HIV and STI testing and treatment/prevention services. What was the rationale for developing an HIV risk assessment tool? And how is it anticipated that such a tool could improve their access to HIV and STI services?

At times in the manuscript, there is mention of the tool being about HIV prevention; other times it states that the tool would help improve access to HIV testing, prevention and treatment. Relating to the point above this, it is not clear what the app is intending to achieve and how this tool expected to contribute to HIV prevention? This needs clarification by clarifying what information the tool is intended to give young. 

Overall the discussion needs to reflect more critically on the study findings, and how these relate the published literature. For example, have other studies explored young people’s understanding of questions included in sexual behaviour questionnaires? How have other studies engaged youth in the development of interventions for youth – there is a growing body of literature on youth engagement that would strengthen the discussion. What other studies have used mHealth to improve access to HIV/STI services among young people, and how? 

Many of the references included in the manuscript are quite old, related to the point above, there is a need to include more recent evidence in the paper. 

Abstract: 

• Change HIV/AIDS to HIV epidemic

• Specify age when referring to young people, and include prevention here – as the following sentence is about barriers to accessing HIV prevention services 

• The sentence “Barriers to confidentiality …” needs rephrasing. I think the sentence aims to say that confidentiality is a concern among young people and thus acts as a barrier to service access, and that online services could address this barrier – but it’s not clear if that’s correct from how the sentence is currently written 

• “Using a qualitative research design..” remove the word design from here

• It’s not clear what is meant by “we conducted a cross-sectional exploratory study of young key populations” consider rephrasing this

• LGBT individuals among other populations are mentioned in the abstract – is the study focussed on young key populations? If so, this needs to be made clear earlier on in the abstract and in the title

• “…mobile phone usage questionnaire..” is not clear, this needs rephrasing 

• The abstract should include details on how many participants were included in the study

Author Summary 

• The sentence starting “Facilities…” needs some rephrasing 

Introduction 

• The introduction needs to focus more on the risk of HIV among young key populations, as the target population for this study; 

• The sentence “Young South Africans are predominantly…” needs some rephrasing, not clear what is meant by this – perhaps that they are at increased risk of HIV?

• It’s not clear how an HIV risk assessment tool would address barriers to confidentiality (long queues, sharing facilities with adults etc) among young people – this needs clarification, i.e. how would a mobile-phone based risk assessment tool would improve service access without addressing the barriers highlighted? 

• The sentence “…while many app have been developed for prevention…” needs references and please include examples of available app

• The last paragraph on page 5 is more appropriate for the methods

• The final paragraph should describe the aim of the study, ie. The second paragraph of page 6 could be rephrased into the aim and added to the end of the last paragraph of the introduction 

Methods

• Spell out the acronyms included in the methods (e.g. PGD)

• Remove the term cross sectional – this is reserved for quantitative studies

• Have the first paragraph be study population and location, then the second is study design

• Remove the word “massive”, rather give an indication of how many patients the walk-in clinic sees daily (for example)

• The acronym HCT is not commonly used anymore, revise to HIV testing services (HTS)

• The first sentence under Recruitment process needs some rephrasing – move participated in the study before breaking down participation by each data collection method

• Under the IDI participants heading, avoid the terms: “hard-to-reach” instead use marginalized, hidden, or minoritized; avoid “subjects”, instead use individuals and/or study participants; instead of access consider reach lesbian, gay etc 

o It says here young people were asked to recruit other people who met the eligibility criteria – but these should be included under study population as it’s not clear until this section that young key populations are among the study population 

• On page 8, under peer group discussions, it’s not clear what was included in the 10 minute risk assessment questionnaire; also it is not clear what “assessing for other STIs..” means? Perhaps self-reported ever having had an STI? This needs clarification, also which STIs? Or were they asked about any STI (other than HIV)

• Page 9, first paragraph – this is the first time it is stated that the risk assessment will also assess for risk of STIs other than HIV, this should be made clearer in the introduction (i.e. that the risk assessments is for STIs, including HIV)

o In this same paragraph, why were participants asked to complete this as a group?

o The labelling on figure 1 (and later figures) need to be clearer (larger font, no acronyms etc) 

• Under in-depth interviews, the term one-off is not necessary here

• Page 10, you can remove the text in brackets reminding the reader who the key populations are

• Considering this is a qualitative study, the measurement section is not necessary particularly as this section is not about measurement but about topics explored. Rather this information could be integrated in the data collection section 

• The data analysis section does not need two headings (ie. Don’t need data analysis and qualitative data analysis) 

o It says here: “All codes were discussed, and any discrepancies were resolved.” But earlier it states that only one person coded the data, what discrepancies are meant in this sentence?

o Consider revising the sentence: “Descriptive statistics and frequencies were used to analyze socio-demographic characteristics and mobile phone usage data…” to Socio-demographic and mobile phone usage data were described. …

Results 

• The first paragraph mentions that 2 individuals were transgender – were they transmen or transwomen or both? 

• The last sentence of the first paragraph states: Overall, 86% (30/35) had access to the internet via a mobile phone, tablet, laptop, or computer. By saying overall the reader assumes among all study participants – the denominator should be the 40 individuals, with the % changed accordingly.

• Although quotes are included in the table, there should be additional quotes included in the text to support the assertions made in the text – for the PGD and the IDI

• Statements such as: All questions considered problematic were revised in collaboration with participants and discussed further with the research team for corroboration. Questions approved by participants and the research team were then sent to the software developer to incorporate into the HRC. Should be moved to the discussion – the results are to present study findings only. 

Discussion 

• The first paragraph of the discussion needs to restate the key findings of the study

• In the second paragraph of the discussion, are there any studies that explored young people’s understanding of questions included in survey questionnaires? 

o This paragraph states that the study expands on the limited evidence of the impact of mobile phones on risk assessment but you study does explore impact, rather it engages young people in the design of the app 

• In the conclusion, it would be useful to state whether the tool will be evaluated and, if so, how?

Reviewer #2: This manuscript focuses on an important health disparity and makes an important contribution to the literature by documenting the process used to engage young people in the development of an mHealth health promotion tool. I have a few suggestions, mainly to enhance the clarity of the information presented.

1) Populations of focus: The Abstract and Author Summary mention persons who inject drugs as a population of focus for this study (along with LGBT individuals, MSM, and female sex workers); however, the Methods and Results in the main text do not describe the specific recruitment of persons who inject drugs for the study. In addition, in some places LGBT individuals and MSM are presented as separate populations and in others (e.g., the subsection titled "Semi-structured interview guides") it seems that these two populations may have been combined in to a single LGBT category. These aspects could be made clearer/more consistent.

2) Data collection: It would be helpful if the authors could explain in the Methods section the languages used in the discussions/interviews vs. the written assessment. This is explained later (e.g., in the Limitations section), but it would be clearer if stated more explicitly when first describing data collection. In addition, were both the demographic and mobile phone usage questionnaire and the HIV risk assessment questionnaire paper-based? As written, the risk assessment questionnaire is described as paper-based but the format for the demographic/mobile phone usage questionnaire is not specified.

3) Demographics: Given that sex workers were one of the populations specifically recruited for the study, and they are a population traditionally considered difficult to reach by researchers, it would be valuable, if possible, if the authors could include in the demographics section the number of participants who self-identified as sex workers.

4) Assessment development process: I was left with two main questions about the process used to develop the HIV risk assessment tool. First, regarding the question on the assessment about race and participants' concerns about it further stigmatizing specific groups: the authors state that the decision was made to leave this question on the assessment and to provide an explanation about its purpose to the young people. Was this explanation just provided to PDG participants, or was an explanation embedded in the assessment fo future users to read when they get to that question? Second, more information about the transition to a chatbot format would be helpful. Is the chatbot an app (vs. a website), and/or how does it address the concerns participants brought up about data limitations?

5) Tables: Some of the abbreviations in the Tables (e.g., IDIX and IDI WOW) were not clear to me. It would be helpful if these could be defined. In addition, in some cases the participant age is included with the quote, in others the date, and in others neither. I recommend making this format more consistent, if possible.

6) Terminology: For consistency throughout the manuscript and to use the most up-to-date language, in the subsection titled "Semi-structured interview guides" I suggest changing "LGBTs" to "LGBT indviduals", and changing "homosexual" in Results to "lesbian or gay".

Again, both the development process and intervention presented in this manuscript are innovative and of interest to those both in research and practice. My main suggestion is to clarify the points above.

6. PLOS authors have the option to publish the peer review history of their article (what does this mean?). If published, this will include your full peer review and any attached files.

**Do you want your identity to be public for this peer review?** For information about this choice, including consent withdrawal, please see our Privacy Policy.

Reviewer #1: No

Reviewer #2: No

---

## [Decision Letter · Decision Letter 1]

28 Nov 2023

PDIG-D-23-00013R1

Developing a youth-friendly internet-enabled HIV risk calculator: A collaborative approach with young people including key population aged 18–24 years living in Soweto, South Africa”.

PLOS Digital Health

Dear Dr. Mulaudzi,

Thank you for submitting your manuscript to PLOS Digital Health. After careful consideration, we feel that it has merit but does not fully meet PLOS Digital Health's publication criteria as it currently stands. Therefore, we invite you to submit a revised version of the manuscript that addresses the points raised during the review process.

Please submit your revised manuscript within 60 days Jan 27 2024 11:59PM. If you will need more time than this to complete your revisions, please reply to this message or contact the journal office at digitalhealth@plos.org. Please include the following items when submitting your revised manuscript:

We look forward to receiving your revised manuscript.

Kind regards,

Padmanesan Narasimhan, MBBS MPH PhD

Section Editor

PLOS Digital Health

Journal Requirements:

Additional Editor Comments (if provided):

Reviewers' comments:

Reviewer's Responses to Questions

**Comments to the Author**

1. If the authors have adequately addressed your comments raised in a previous round of review and you feel that this manuscript is now acceptable for publication, you may indicate that here to bypass the “Comments to the Author” section, enter your conflict of interest statement in the “Confidential to Editor” section, and submit your "Accept" recommendation.

Reviewer #2: All comments have been addressed

2. Does this manuscript meet PLOS Digital Health’s publication criteria? Is the manuscript technically sound, and do the data support the conclusions? The manuscript must describe methodologically and ethically rigorous research with conclusions that are appropriately drawn based on the data presented.

Reviewer #2: Yes

3. Has the statistical analysis been performed appropriately and rigorously?

Reviewer #2: Yes

4. Have the authors made all data underlying the findings in their manuscript fully available (please refer to the Data Availability Statement at the start of the manuscript PDF file)?

Reviewer #2: Yes

5. Is the manuscript presented in an intelligible fashion and written in standard English?

Reviewer #2: Yes

6. Review Comments to the Author

Reviewer #2: The authors have thoroughly addressed the reviewer comments, which has helped with clarity. I suggest reviewing for minor edits, such as to make sure that all acronyms are defined at first use (for example, the acronyms PGDs and IDIs are used on page 9 of the tracked changes version under "Study population and setting", but not defined until later in the manuscript). There are also some participant quotations that are repeated in both Table 3 and in the main text; these could be removed from one of the locations to reduce repetition. Finally, when reporting participant demographics (e.g., in Table 1 and in the subsection titled "Participant demographics and mobile phone usage"), I recommend using the terms "cisgender male" and "cisgender female" (as the counterparts to "transgender" or "transgender female") rather than "male" and "female" alone.

7. PLOS authors have the option to publish the peer review history of their article (what does this mean?). If published, this will include your full peer review and any attached files.

**Do you want your identity to be public for this peer review?** For information about this choice, including consent withdrawal, please see our Privacy Policy. 

Reviewer #2: No

---

## [Decision Letter · Decision Letter 2]

29 Dec 2023

PDIG-D-23-00013R2

Developing a youth-friendly internet-enabled HIV risk calculator: A collaborative approach with young people including key population aged 18–24 years living in Soweto, South Africa”.

PLOS Digital Health

Dear Dr. Mulaudzi,

Thank you for submitting your manuscript to PLOS Digital Health. After careful consideration, we feel that it has merit but does not fully meet PLOS Digital Health's publication criteria as it currently stands. Therefore, we invite you to submit a revised version of the manuscript that addresses the points raised during the review process.

Please submit your revised manuscript within 60 days Feb 27 2024 11:59PM. If you will need more time than this to complete your revisions, please reply to this message or contact the journal office at digitalhealth@plos.org. Please include the following items when submitting your revised manuscript:

We look forward to receiving your revised manuscript.

Kind regards,

Haleh Ayatollahi

Section Editor

PLOS Digital Health

Journal Requirements:

Additional Editor Comments (if provided):

Please follow the journal instructions for organizing different parts of the manuscript including the abstract. Moreover, I think the title is a bit long, please make it shorter.

Reviewers' comments:

Reviewer's Responses to Questions

**Comments to the Author**

1. If the authors have adequately addressed your comments raised in a previous round of review and you feel that this manuscript is now acceptable for publication, you may indicate that here to bypass the “Comments to the Author” section, enter your conflict of interest statement in the “Confidential to Editor” section, and submit your "Accept" recommendation.

Reviewer #3: All comments have been addressed

Reviewer #4: (No Response)

Reviewer #5: (No Response)

Reviewer #6: All comments have been addressed

2. Does this manuscript meet PLOS Digital Health’s publication criteria? Is the manuscript technically sound, and do the data support the conclusions? The manuscript must describe methodologically and ethically rigorous research with conclusions that are appropriately drawn based on the data presented.

Reviewer #3: Yes

Reviewer #4: Partly

Reviewer #5: Partly

Reviewer #6: Yes

3. Has the statistical analysis been performed appropriately and rigorously?

Reviewer #3: I don't know

Reviewer #4: N/A

Reviewer #5: N/A

Reviewer #6: I don't know

4. Have the authors made all data underlying the findings in their manuscript fully available (please refer to the Data Availability Statement at the start of the manuscript PDF file)?

Reviewer #3: Yes

Reviewer #4: Yes

Reviewer #5: No

Reviewer #6: No

5. Is the manuscript presented in an intelligible fashion and written in standard English?

Reviewer #3: Yes

Reviewer #4: Yes

Reviewer #5: No

Reviewer #6: Yes

6. Review Comments to the Author

Reviewer #3: The research article, "Developing a youth-friendly internet-enabled HIV risk calculator: A collaborative approach with young people including key population aged 18–24 years living in Soweto, South Africa," focuses on creating an HIV risk tool for youth. 

It conducts a qualitative study with young people in Soweto, South Africa, aiming to create an accessible and interactive tool. 

The manuscript is well-structured, addressing a crucial health issue, but it needs more precise language in methodology and results. It should explicitly discuss how it fills literature gaps, particularly in mHealth for HIV prevention. 

The paper commendably engages with youth but should delve deeper into the tool's real-world impact, addressing implementation challenges. While it presents data clearly, incorporating more visual aids could enhance understanding. Ethical considerations are well-handled, yet further discussion on the implications of involving young populations would strengthen it. 

The manuscript briefly mentions future research but could expand on this, discussing potential scalability. 

Minor revisions could significantly enhance its clarity and expand its impact, making it a valuable field contribution.

Reviewer #4: (No Response)

Reviewer #5: 1. Overall:

1.1. I found this manuscript to be very insightful, and a good example of following the principles for digital development, notably “design with the user”, and “address privacy”. These findings could indeed be helpful for other groups interested in developing similar tools.

Minor Comments:

2. ABSTRACT

2.1. Line 29: Would consider avoid using colloquial/informal terms such as “tap”

2.2. HIV risk calculator (HRC): if the editor agrees, I would avoid using the acronym HRC as it is not commonly used, and thus difficult to understand for readers that may have skipped a section.

2.3. Typo/grammar line 31: unnecessary “the”

2.4. “young key populationS groups” please correct to “young key population groups”

2.5. Line 34 to 38: Very long sentence, consider splitting 

2.6. Line 40: sentence structure confusing

2.7. Consider outlining in abstract why the selected groups are “key populations”, I assume because they are groups at highest risk of HIV infection, or rather because they are less likely to access HIV care?

2.8. Conclusion: “Privacy, confidentiality, and ease of use promote acceptability and willingness to use internet-enabled HIV prevention methods.” Should be reworded to be less assertive. Consider using terms such as “Participants found/emphasized/highlighted” etc. 

3. Author summary:

3.1. Line 60: “Facilities are available” reword

3.2. Line 61: Not clear why “long gueues at testing facilities and lack of youth-specific health care services” are barriers to confidentiality. Please explain or modify sentence.

3.3. Lack of confidentiality jeopardizes young people’s futures and lives. Is a big statement. Is this what was meant?

3.4. No mention about how the HIV risk calculator helps youth assess their risk of HIV infection (is this not the main goal?).

4. Introduction

4.1. Line 78: consider rewording

4.2. Line 91: “serious worry” consider rewording

5. Methods

5.1. Line 141 typo

5.2. Line 169: “They could contact the study team as a walk-in” not clear what that means.

5.3. Line 209 typo

5.4. Line 215:” IDIs concluded with a discussion around the HRC” what was discussed “around the HRC”?

5.5. Line 286: Would suggest providing equivalent amount in dollars or euros in parenthesis beside R150, for reader’s outside of South Africa to understand

6. Discussion

“The HIV risk calculator was evaluated with young South Africans as part of a PhD degree and results are available in the form of a thesis on the database of the University of the Witwatersrand in Johannesburg South Africa. “ Would remove this.

6.1. 

Major Comments

1. There are many grammatical and language structure errors in some sections of the manuscript (abstract in particular) that would need to be corrected before it is ready to publish. Some were highlighted in this review, but many were not.

2. Methods: More details would be required to understand the IDI “purposive sampling” approach. From the way it reads, it sounds like a convenience sample using a chain referral strategy including participants who met the study’s inclusion criteria.

3. Methods:Line 201-204: “After completing the HRC prototype” Does this mean a third round of PGD was held? Please clarify including the necessary information to understand the modality of this PGD, or reword to clarify what this means.

4. Results: At no point in the manuscript does it outline how the HIV risk calculator works, in the spirit of the principles for digital development, and for other groups to better understand the tool, it would be important to share how the calculator works.

5. The discussion could benefit from a mention on the impact of the modifications made to the digital app through the collaborative development process. Potential for improved uptake? Improved accuracy of the risk calculator?

6. A framework such as the principles for digital development should be mentioned to highlight what aspects were followed, and highlight which aspects were not followed as limitations.

Reviewer #6: Mamakiri Mulaudzi and colleagues have revised their manuscript twice following reviewer suggestions. They have adequately addressed all concerns.

7. PLOS authors have the option to publish the peer review history of their article (what does this mean?). If published, this will include your full peer review and any attached files.

**Do you want your identity to be public for this peer review?** For information about this choice, including consent withdrawal, please see our Privacy Policy. 

Reviewer #3: Yes: Dr K Madan Gopal

Reviewer #4: No

Reviewer #5: Yes: Rainer Tan

Reviewer #6: No

---

## [Decision Letter · Decision Letter 3]

26 Jun 2024

PDIG-D-23-00013R3

Developing a youth-friendly internet-enabled HIV risk calculator: A collaborative approach with young key population aged 18–24 years living in Soweto, South Africa”.

PLOS Digital Health

Dear Dr. Mulaudzi,

Thank you for submitting your manuscript to PLOS Digital Health. After careful consideration, we feel that it has merit but does not fully meet PLOS Digital Health's publication criteria as it currently stands. Therefore, we invite you to submit a revised version of the manuscript that addresses the points raised during the review process.

Please submit your revised manuscript within 60 days Aug 25 2024 11:59PM. If you will need more time than this to complete your revisions, please reply to this message or contact the journal office at digitalhealth@plos.org. Please include the following items when submitting your revised manuscript:

We look forward to receiving your revised manuscript.

Kind regards,

Haleh Ayatollahi

Section Editor

PLOS Digital Health

Journal Requirements:

Additional Editor Comments (if provided):

Reviewers' comments:

Reviewer's Responses to Questions

**Comments to the Author**

1. If the authors have adequately addressed your comments raised in a previous round of review and you feel that this manuscript is now acceptable for publication, you may indicate that here to bypass the “Comments to the Author” section, enter your conflict of interest statement in the “Confidential to Editor” section, and submit your "Accept" recommendation.

Reviewer #3: All comments have been addressed

Reviewer #5: (No Response)

Reviewer #6: All comments have been addressed

2. Does this manuscript meet PLOS Digital Health’s publication criteria? Is the manuscript technically sound, and do the data support the conclusions? The manuscript must describe methodologically and ethically rigorous research with conclusions that are appropriately drawn based on the data presented.

Reviewer #3: Yes

Reviewer #5: Partly

Reviewer #6: Yes

3. Has the statistical analysis been performed appropriately and rigorously?

Reviewer #3: Yes

Reviewer #5: N/A

Reviewer #6: I don't know

4. Have the authors made all data underlying the findings in their manuscript fully available (please refer to the Data Availability Statement at the start of the manuscript PDF file)?

Reviewer #3: Yes

Reviewer #5: No

Reviewer #6: Yes

5. Is the manuscript presented in an intelligible fashion and written in standard English?

Reviewer #3: Yes

Reviewer #5: No

Reviewer #6: Yes

6. Review Comments to the Author

Reviewer #3: no comments.

Reviewer #5: None of the proposed clarifications and questions put forward by reviewer 5 during the second revision were addressed in the present answer by the authors. Please address these points.

Reviewer #6: All comments have been addressed.

7. PLOS authors have the option to publish the peer review history of their article (what does this mean?). If published, this will include your full peer review and any attached files.

**Do you want your identity to be public for this peer review?** For information about this choice, including consent withdrawal, please see our Privacy Policy. 

Reviewer #3: Yes: Dr K Madan Gopal

Reviewer #5: Yes: Rainer Tan

Reviewer #6: Yes: Pinkus Tober-Lau

---

## [Decision Letter · Decision Letter 4]

18 Oct 2024

Developing a youth-friendly internet-enabled HIV risk calculator: A collaborative approach with young key population aged 18–24 years living in Soweto, South Africa”.

PDIG-D-23-00013R4

Dear Dr Mulaudzi,

We are pleased to inform you that your manuscript 'Developing a youth-friendly internet-enabled HIV risk calculator: A collaborative approach with young key population aged 18–24 years living in Soweto, South Africa”.' has been provisionally accepted for publication in PLOS Digital Health.

Best regards,

Haleh Ayatollahi

Section Editor

PLOS Digital Health

Reviewer Comments (if any, and for reference):

Reviewer's Responses to Questions

**Comments to the Author**

1. If the authors have adequately addressed your comments raised in a previous round of review and you feel that this manuscript is now acceptable for publication, you may indicate that here to bypass the “Comments to the Author” section, enter your conflict of interest statement in the “Confidential to Editor” section, and submit your "Accept" recommendation.

Reviewer #5: All comments have been addressed

2. Does this manuscript meet PLOS Digital Health’s publication criteria? Is the manuscript technically sound, and do the data support the conclusions? The manuscript must describe methodologically and ethically rigorous research with conclusions that are appropriately drawn based on the data presented.

Reviewer #5: Yes

3. Has the statistical analysis been performed appropriately and rigorously?

Reviewer #5: N/A

4. Have the authors made all data underlying the findings in their manuscript fully available (please refer to the Data Availability Statement at the start of the manuscript PDF file)?

Reviewer #5: Yes

5. Is the manuscript presented in an intelligible fashion and written in standard English?

Reviewer #5: No

6. Review Comments to the Author

Reviewer #5: All my comments have been responded to. Congrats to the authors for the many revisions, the latest manuscript, and importantly the HIV risk tool.

Of note there is a spelling error in the title: “young key populations” not “young key population” Could also consider dropping “aged 18-24 years” to shorten the title (but not mandatory). There remains other spelling errors and typos throughout the manuscript (ex. Line 220), so could benefit from a last careful review of spelling and typos.

7. PLOS authors have the option to publish the peer review history of their article (what does this mean?). If published, this will include your full peer review and any attached files.

**Do you want your identity to be public for this peer review?** For information about this choice, including consent withdrawal, please see our Privacy Policy.

Reviewer #5: **Yes: **Rainer Tan
